# The Potential of Telomeric G-Quadruplexes Containing Modified Oligoguanosine Overhangs in Activation of Bacterial Phagocytosis and Leukotriene Synthesis in Human Neutrophils

**DOI:** 10.3390/biom10020249

**Published:** 2020-02-06

**Authors:** Ekaterina A. Golenkina, Galina M. Viryasova, Nina G. Dolinnaya, Valeria A. Bannikova, Tatjana V. Gaponova, Yulia M. Romanova, Galina F. Sud’ina

**Affiliations:** 1Lomonosov Moscow State University, Belozersky Institute of Physico-Chemical Biology, Moscow 119234, Russia; golyesha@mail.ru (E.A.G.); gali-inimitable@yandex.ru (G.M.V.); 2Lomonosov Moscow State University, Department of Chemistry, Moscow 119234, Russia; dolinnaya@hotmail.com (N.G.D.); bannikova.valeria@yandex.ru (V.A.B.); 3National Research Center for Hematology, Russia Federation Ministry of Public Health, Moscow 125167, Russia; gaponova.tatj@yandex.ru; 4Gamaleya National Research Centre of Epidemiology and Microbiology, Moscow 123098, Russia; genes2007@yandex.ru

**Keywords:** neutrophils, phagocytosis, leukotrienes, oligodeoxyribonucleotides, G-quadruplexes, salmonella

## Abstract

Human neutrophils are the first line of defense against bacterial and viral infections. They eliminate pathogens through phagocytosis, which activate the 5-lipoxygenase (5-LOX) pathway resulting in synthesis of leukotrienes. Using HPLC analysis, flow cytometry, and other biochemical methods, we studied the effect of synthetic oligodeoxyribonucleotides (ODNs) able to fold into G-quadruplex structures on the main functions of neutrophils. Designed ODNs contained four human telomere TTAGGG repeats (G4) including those with phosphorothioate oligoguanosines attached to the end(s) of G-quadruplex core. Just modified analogues of G4 was shown to more actively than parent ODN penetrate into cells, improve phagocytosis of *Salmonella typhimurium* bacteria, affect 5-LOX activation, the cytosol calcium ion level, and the oxidative status of neutrophils. As evident from CD and UV spectroscopy data, the presence of oligoguanosines flanking G4 sequence leads to dramatic changes in G-quadruplex topology. While G4 folds into a single antiparallel structure, two main folded forms have been identified in solutions of modified ODNs: antiparallel and dominant, more stable parallel. Thus, both the secondary structure of ODNs and their ability to penetrate into the cytoplasm of cells are important for the activation of neutrophil cellular effects. Our results offer new clues for understanding the role of G-quadruplex ligands in regulation of integral cellular processes and for creating the antimicrobial agents of a new generation.

## 1. Introduction

Studying the effects of synthetic small molecules on human polymorphonuclear leukocytes (PMNLs or neutrophils) via cellular receptors is currently considered as a promising approach for both the regulation of cellular functions and developing new antibacterial drugs. Toll-like receptors (TLRs) are known to recognize microbes by sensing pathogen-associated molecular patterns (PAMPs) [1] and initiate signaling ways leading to inflammatory response [1,2]. Recently, particular attention has been paid to TLR ligands, which have a modulating effect on innate immunity and are therefore promising agents for prevention and treatment of infectious diseases. Agonist of TLR2 receptor has been shown to improve microbial clearance at bacterial infection [3], and TLR4 agonist stimulated bacterial clearance in *Pseudomonas aeruginosa* infection [4]. However, the most perspective target for suppression of viral and bacterial infections can be TLR9. Ligand binding to TLR9 receptor activates NF-κB-mediated release of inflammatory mediators [5]. TLR9 receptor, known as a microbial DNA sensor, recognizes DNA fragments containing unmethylated d(CpG) motifs (CpG-DNA), which are abounded in bacterial unlike in eukaryotic DNA [6]. After binding to TLR9, synthetic oligodeoxyribonucleotides (ODNs) containing d(CpG) sequences (CpG-ODNs) of different classes (agonistic ligands), which mimic bacterial degradation products, activate receptor and initiate an inflammatory response. Activation of TLR9 triggers the signaling cascade leading to the production of antimicrobial products and proinflammatory cytokines [7]. The coordinated action of all participants of inflammatory response can lead to clinical signs of sepsis and septic shock [8]. To suppress prolonged inflammation, it seems appropriate to use the receptor inhibitory ODNs, which allow negative regulation of TLR9 signaling [9]. It was shown that telomeric TTAGGG tandem repeats in synthetic ODNs can selectively block the colocalization of CpG-DNA with TLR9 [10], as well as CpG-mediated immune activation [11] acting as TLR9 ligands [12].

Innate immunity rapidly generates an inflammatory response to PAMPs by Toll-like receptors. The biological activity of CpG-ODNs and putative ways for its directed regulation by inhibitory G-rich ODNs are shown schematically in Scheme 1.

The spectrum of possibilities for therapeutic use of TLR9 oligonucleotide ligands is expanding, since expression of this receptor is a prognostic criterion for viral diseases leading to immunosuppression [13,14]. After the immunoregulatory role of bacterial DNA has been proven, the use of ODNs mimicking the receptor-recognized sites of bacterial DNA as stimulators of innate immunity seems to be an attractive alternative to vaccination and antibiotic therapy. TLR9 is also widely expressed in tumor cells [15], and its agonists—CpG-ODNs—promote cancer cell invasion [16,17]. Synthetic oligonucleotides containing TTAGGG repeats of humane telomere DNA also induce TLR9-mediated invasion of breast cancer cells [12]. The invasive effects of ODNs are regulated by LL-37 peptide present in breast cancer cells. Complexing with the LL-37 peptide enhances the cellular uptake of oligonucleotides and reduces cell invasion when the complex with the peptide forms an ODN with telomeric repeats rather than CpG-ODN [12]. The telomeric repeats in synthetic ODNs can also prevent STAT1 and STAT4 phosphorylation and attenuate a variety of inflammatory responses in vivo through blocking TLR9 signaling way [18,19]. The biological activity of telomeric DNA repeats can be associated with their ability to fold into non-canonical G-quadruplex structures.

G-quadruplexes are formed via intra- or intermolecular interactions in DNA or RNA molecules containing oligoG repeats (so-called G-tracts). The core of the G-quadruplex consists of two or more stacked planar G-tetrads, in which four guanine bases from different G-tracts interact via Hoogsteen bonding, while the intervening sequences are extruded as loops. DNA and RNA G-quadruplexes are the only non-canonical structures whose presence within genomes was strictly proved [20,21]. Being structural elements of the genome, G-quadruplexes are recognized by numerous proteins and enzymes, whose activity is modulated by these non-canonical DNA structures. They affect the most important biological processes, such as replication, chromosome end protection, transcription, translation, mutagenesis, and DNA recombination [22,23,24,25]. Recent studies have shown that G-quadruplexes exist in genome DNA in all phases of the cell cycle. These non-canonical structures are two-faced. On the one hand, they can perform regulatory functions in the cell, inhibit oncogene expression, block undesired elongation of telomere DNA, control the level of negative supercoiling in the genome, and serve as a target for antitumor therapy. On the other hand, G-quadruplex formation causes genome instability (inversions, recombination, mutations, deletions, etc.) associated with oncological diseases and neurological disorders [20].

Recently it was shown that G-quadruplex-forming ODNs activate leukotriene synthesis in human neutrophils [26], though the role of TLR receptors in this process has not yet been proven. Leukotriene B4 (LTB4) is a well-recognized mediator in immune responses to infections [27,28]. LTB4 potentiates microbial phagocytosis and killing [29,30], boosts innate immunity [31,32], and recruits and activates human neutrophils via G-protein-coupled receptors [33,34]. LTB4 is a unique lipid mediator that interacts with both cell-surface and nuclear receptors; LTB4 can also bind and activate transcription factor PPAR (peroxisome proliferator-activated receptor) alpha [35,36], resulting in the activation of genes that terminate inflammatory processes [37].

The aim of this study is to evaluate the potential of ODNs containing human telomeric TTAGGG tandem repeats in activation of bacterial phagocytosis and the leukotriene synthesis in human neutrophils. ODNs are known to have a great capability for various therapeutic applications, and the study of nuclease-resistant DNA quadruplexes will help to find efficient synthetic modulators of immune responses. Here, we designed a set of ODNs containing four human telomeric repeats with natural phosphodiester internucleotide bonds having the ability to fold into intramolecular G-quadruplex. These constructs differ in the presence of phosphorothioate oligoguanosines of varied length attached to the 3′-end or both to 3′- and 5′-ends of G-quadruplex sequence motif (Table 1). The oligoguanosine runs have been shown to significantly increase the cellular uptake and to improve immunostimulating activity of CpG-ODNs targeting TLR9 receptors [38]. Especially efficient was 3′-end flanking with oligoguanosines containing phosphorothioate internucleotide bonds instead of natural phosphodiester ones [39]. In this study, we estimated, using circular dichroism and UV spectroscopy, the folding topology and thermal stability of G-quadruplexes formed by designed ODNs and elucidated how specific patterns of G-quadruplexes can regulate the numerous neutrophil properties. 

## 2. Materials and Methods 

In this study, we used the ODNs synthesized and purified (HPLC and PAGE) by DNA-synthesis (Moscow, Russia). The strand concentrations of ODNs dissolved in the appropriate buffer solutions were determined spectrophotometrically. 

### 2.1. Preparation of Bacteria

*Salmonella typhimurium* (*S*. typhimurium) strain IE 147 was obtained from the Collection of Gamaleya National Research Centre of Epidemiology and Microbiology (Moscow, Russia). Bacteria (stock solution contained 1 × 10^9^ colony-forming units (CFU)/mL) were grown in Luria–Bertani broth, washed twice with a physiological salt solution, and collected by centrifugation at 2000× *g*. Then the bacteria were opsonized for 30 min in 20% (v/v) fresh serum from the same donor whose blood was used to prepare the neutrophils. Bacteria were washed by repeated centrifugation in Dulbecco’s solution.

### 2.2. Isolation of Neutrophils

Human neutrophils were isolated from the blood of healthy volunteers, who provided their informed consent and with ethics approval from the Institutional Ethics Committee of the A. N. Belozersky Institute of Physico-Chemical Biology, Lomonosov Moscow State University. Neutrophils were isolated from freshly drawn citrate-anticoagulated donor peripheral blood by standard techniques, as previously described [40]. Leukocyte-rich plasma was prepared by 3% dextran T-500 sedimentation of erythrocytes at room temperature. Granulocytes were purified by centrifugation through Ficoll–Paque (the density is 1.077 g/mL) followed by hypotonic lysis of the remaining erythrocytes. PMNLs were washed twice with PBS, resuspended at 10^7^/mL (purity of 96–97%, viability of 98–99%) in Dulbecco’s PBS containing 1 mg/mL glucose (without CaCl_2_), and stored at room temperature. 

### 2.3. Phagocytosis Assessment

#### 2.3.1. Preparation of Fluorescein-Labeled and Opsonized Bacteria

To prepare fluorescein-labeled *S*. typhimurium, bacteria (1 × 10^9^ CFU/mL) were suspended in 1 mL of 0.05% FITC (fluorescein-isothiocyanate, isomer I, Merck, Kenilworth, NJ) in 0.1 M NaHCO_3_ (pH 9.0), and incubated overnight at 4 °C in the dark with constant shaking. The bacteria were then collected by centrifugation at 3000× *g*, washed in PBS twice, and opsonized for 30 min with donor’s serum. After that, bacteria were washed, resuspended in PBS and used for assay. 

#### 2.3.2. Examination of Neutrophils Phagocytic Functions

PMNLs (10^6^ cells in 0.5 mL aliquots of Hanks’ solution, pH 7.4) supplemented with 1 µM ODN, except for control samples, were incubated in Eppendorf tubes for 15 min at 37 °C in 5% CO_2_ with continuous stirring. Then, opsonized FITC-labeled *S*. *typhimurium* bacteria were added, so that their ratio to the neutrophils was approximately 20:1, for further 20 min incubation under the same conditions. PMNLs were collected by centrifugation at 200× *g*, resuspended in PBS followed by FACS analysis (excitation of 488 nm, emission of 525 nm) on Cytomics FC 500 Flow Cytometry System (Beckman Coulter, Germany) with CXP software. To differentiate between internalized and surface-bound bacterial cells, trypan blue (TB), at a working concentration of 1 mg/mL, was used for quenching surface fluorescence due to attached bacteria [41]. Thus, each value was analyzed both without and with the addition of TB just before measurement. Both relative amounts of PMNLs involved in phagocytosis and fluorescence intensities reflecting an average number of bacteria captured per cell were estimated.

### 2.4. Adhesion Assessment

PMNLs (10^6^ cells in 0.5 mL aliquots of Hanks′ solution, pH 7.4), supplemented with ODNs in concentrations required, except for control samples, were incubated in fibrinogen pre-coated 24-well plates for 15 min at 37 °C in 5% CO_2_. Then supernatants were carefully removed and wells were washed twice with a warm PBS to remove unattached cells. A quantitative assessment of adhesion was based on *ortho*-phenylenediamine (OPD) oxidation by hydrogen peroxide catalyzed by myeloperoxidase [42]. Hydrogen peroxide (4 mM final concentration) in permeabilizing buffer (67 mM Na_2_HPO_4_, 35 mM citric acid, 0.1% Triton X-100) supplemented with 5.5 mM OPD was added to substrate-bound PMNLs for 5 min [43]. The reaction was stopped by the addition of 1M H_2_SO_4_ followed by recording the absorption values at 492 nm and their comparison with the calibration values.

### 2.5. Synthesis of Leukotrienes, 5-lipoxygenase (5-LOX) Metabolites of Arachidonic Acid, in Human Neutrophils

The cell suspension (2 × 10^7^ PMNLs, reaction volume of 6 mL) was incubated at 37 °C in a 5% CO_2_ incubator for 15 min with 0.5–1.0 µM ODNs. Then, the reactive mixture was treated with *S. typhimurium* (the ratio of bacteria to PMNLs was approximately 20:1) for a further 15 min to stimulate the 5-LOX pathway [44]. The incubation was stopped by addition of an equal volume of methanol at −20 °C with prostaglandin B2 as an internal standard. The samples were stored at −20 °C. The denatured cell suspension was centrifuged yielding supernatants designated as water/methanol extracts.

### 2.6. HPLC Analysis of Water/Methanol Extracts

The water/methanol extracts were purified by solid-phase extraction using Sep-Pak C18 columns (500 mg; Macherey-Nagel, Dueren, Germany), which were conditioned first with methanol, and then with water. The water/methanol extracts were diluted with PBS containing 0.1% formic acid to obtain samples with methanol content not more than 20% and applied to the cartridges. The cartridges were washed with 10 mL water, and then eluted with methanol. First 200 µL of eluent were discarded, eicosanoids were eluted with the next 1.5 mL of methanol. The solvent was evaporated in Speed-Vac concentrator Savant SC-100 (Farmingdale, NY, USA), the residue was reconstituted in 50 µL of methanol/water (2:1). The purified samples were injected into a 5 µm, 250 × 4.6 mm Nucleosil^®^ C18 column (Macherey-Nagel GmbH) and subjected to RP HPLC. The columns were eluted as described [45]. Products of the 5-LOX pathway were identified by their co-elution with authentic standards; we also proved retention times of the arachidonic acid (AA) metabolites by simultaneous detection for UV absorbance and radioactivity, when added [^14^C]AA to neutrophils [43,46]. The compounds were quantified by comparison of peak areas with the internal standard, prostaglandin B2 [45].

### 2.7. Calcium Ions Influx Assessment

To detect intracellular Ca^2+^, Fluo-3, which is known to dramatically increase fluorescence after calcium ions binding, was used. In accordance with the manufacturer’s protocol, human neutrophils were incubated with 5 μM Fluo-3 AM ester (ThermoScientific, USA) for 60 min at room temperature followed by washing with PBS. After loading, PMNLs were incubated according to the experimental protocol with or without additives in HBSS/HEPES medium. For kinetics assays, PMNLs were seeded in fibrinogen-coated 96-well plates (0.5 × 10^6^/mL HBSS/HEPES), then 1 µM oligonucleotides were added. Changes in Fluo-3-Ca-fluorescense intensities some time before and at least 20 min after ODN addition were recorded on a fluorometric plate reader upon excitation of 485 nm and emission of 538 nm for 20 min at 37 °C. Flow cytometry was also used. In this case, cells (1 × 10^6^/mL HBSS/HEPES medium) were incubated with 1 µM oligonucleotides for 20 min, with or without the addition of opsonized *S. typhimurium* (OS) for the last 10 min of incubation. At the end of the incubation, the samples were placed on ice followed by immediate acquisition on Cytomics FC 500 Flow Cytometry System. Fluorescence was collected by photomultipliers at 525 nm.

### 2.8. Reactive Oxygen Species (ROS) Formation Assay

Intracellular reactive oxygen species (ROS) formation was monitored by measuring green fluorescence of oxidized product of H_2_DCF-DA (2’,7’-dichlorodihydrofluorescein diacetate) included into cells, in accordance with the manufacturer’s protocol. Briefly, human neutrophils were incubated with 5 μM carboxy-H2DCF-DA (ThermoScientific, USA) for 60 min at room temperature followed by washing with PBS. Cells were then seeded in fibrinogen-coated 24-well plates (1 × 10^6^/mL HBSS/HEPES) and incubated according to the experimental protocol for 60 min at 37 °C in 5% CO_2_. Changes in fluorescence intensity were recorded on a fluorometric plate reader upon excitation of 485 nm and emission of 538 nm.

### 2.9. Uptake Assay with FAM-Labeled ODNs

To examine cellular binding and uptake, the oligonucleotides labeled with 6-fluorescein amidite (FAM) at theirs 5′ ends were used. FAM-labeled ODNs (DNA-synthesis, Moscow, Russia) were added to cell suspension (1 × 10^6^/mL HBSS/HEPES) at a concentration of 0.5 µM for 30 min at 37 °C. After incubation, samples were placed on ice, topped with ice-cold 0.1% BSA in PBS followed by centrifugation at 270× *g*. Collected PMNLs were resuspended and analyzed on Cytomics FC 500 Flow Cytometry System (excitation of 488 nm and emission of 525 nm). To distinguish between internalized and surface-bound ODNs, TB in working concentration of 1 mg/mL was used for quenching surface FAM fluorescence.

### 2.10. Circular Dichroism Measurements

For spectroscopic measurements, the ODNs were dissolved in 20 mM HEPES-KOH, 140 mM NaCl, and 5 mM KCl, pH 7.3 (buffer A); water for spectroscopic solutions was obtained from a filtration system (18.2 mΩ). To allow secondary structure formation, each DNA sample was heated to 95 °C for 2–3 min and then cooled overnight to room temperature (annealing process). The CD spectra were recorded in 1 cm path length quartz cells at 30 °C in the wavelength range of 220–360 nm on a Chirascan CD spectrometer (Applied Photophysics Ltd., Leatherhead, UK) equipped with a thermally controlled holder. Measurements were performed at a bandwidth of 1 nm and scanning speed of 30 nm/min with a response time of 2 s and constant flow of dry nitrogen. All the CD spectra were baseline-corrected for signal contributions caused by the buffer solution. CD spectra were presented in units of molar circular dichroism, Δε (cm^–1^ × M^–1^), counting per oligonucleotide vs. wavelength. The oligonucleotide concentration was chosen to give absorption of 0.6–0.8 at 260 nm, which gives an optimum signal-to-noise ratio. CD data represent three averaged scans. The spectra were processed with the Origin 8.0 software using the Savitsky–Golay filter.

### 2.11. UV Spectroscopy Melting Curves

The melting temperature (Tm) measurements of ODNs dissolved and annealed in buffer A were performed on a double-beam Hitachi U-2900 UV/visible spectrophotometer equipped with a Hitachi thermoelectric controller. UV absorption was monitored at 295 nm as a function of temperature using 300 μl quartz microcell (Hellma Analytics, Müllheim, Germany) with 10 mm optical path length. Conformation changes were registered between 10 and 80 °C with heating rate of 0.5 °C/min. The experiments were repeated three times and averaged Tm values were established. The ODN concentration was ~3 µM counting per oligonucleotide.

### 2.12. Statistical Analysis

Data are expressed as mean ± SEM and were analyzed by ANOVA. Data were compared using Tukey’s or Sidak’s multiple comparisons test, where differences were considered significant at *p* < 0.05. GraphPad Prism 8 was used to generate all graphs and conduct statistical analysis.

## 3. Results

In this study, we used synthetic ODNs consisting of four telomeric TTAGGG repeats including those with modified oligoguanosines of different lengths attached to the ends of a G-quadruplex-forming motif. ODN sequences and abbreviations are listed in Table 1. ODN 4084-F, which is known as TLR9 antagonist [47] and neutralizer of CpG-ODN stimulatory activity [48], was also investigated; all internucleotide bonds in ODN 4084-F are modified phosphorothioate ones. The ODN 4084-F was taken for investigation as the shortest G-rich oligonucleotide binding with inhibitory locus of TLR9 receptor [47].

### 3.1. ODNs with Four Telomeric TTAGGG Repeats Flanked with Modified Oligoguanosines Enhance Phagocytic Activity and Adhesiveness of Neutrophils

Human polymorphonuclear leukocytes synthesize the leukotrienes in response to diverse stimuli such as phagocytosis of opsonized targets or soluble stimuli. We investigated leukotriene synthesis by neutrophils after cell stimulation by *S*. *typhimurium* bacteria. The formation of reaction products catalyzed by 5-LOX correlates with phagocytic activity of neutrophils [49]. Phagocytosis is crucial in the fight against infections, ensuring the capture and digestion of bacterial cells and triggering the synthesis of bactericidal and immunomodulating agents.

To begin, we estimated the OND potential to affect phagocytosis of *S*. *typhimurium* bacteria by neutrophils. According to the data obtained, the presence of non-modified ODN G4 did not affect PMNLs phagocytic activity (Figure 1). At the same time, the ODNs containing phosphorothioate oligoguanosines attached to the ends of G-quadruplex-forming motif (G4-g_5_ and g_2_-G4-g_5_) clearly intensified the process of phagocytosis, increasing the proportion of neutrophils that interact with bacterial cells and positively affect the phagocytosis completeness (Figure 1a,c). The relative extent of bacteria capturing also increased markedly under conditions of short-term preliminary incubation of neutrophils with ODNs containing four telomere repeats modified at the ends (Figure 1b,c). 

The signaling mechanisms that trigger the activation of neutrophils during substrate adhesion are close to those induced by phagocytosis [50]. Being supplemented with the non-modified ODN G4, neutrophil adhesiveness increased insignificantly. The concentration-dependent effect of the modified ODNs with terminal phosphorothioate oligoguanosine runs was much higher and was already observed at such low concentration as 0.1 µM (data not shown). So, 1 µM ODN G4-g_5_ doubled the fraction of adherent cells, while ODN g_2_-G4-g_5_ at the same concentration tripled the adhesiveness compared to the control sample without the addition of an oligonucleotide. The efficiency of cell adhesion to the substrate induced by g_2_-G4-g_5_ even exceeded that under the action of the TLR9 antagonist ODN 4084-F (Figure 2).

### 3.2. Priming with ODNs Potentiates the Synthesis of Leukotrienes during the Phagocytosis of Opsonized S. Typhimurium Bacteria by Human Neutrophils

Recently we have shown that G-quadruplex-forming ODNs containing mammalian telomere repeats enhanced leukotriene synthesis in human neutrophils [26]. Here, 5-LOX products were detected after 15 min priming of human neutrophils with 0.5 or 1.0 µM ODN followed by 15 min stimulation with opsonized *S*. *typhimurium* bacteria. Although the addition of oligonucleotides did not cause any detectable formation of 5-LOX products in the absence of bacteria as targets for phagocytosis, the synthesis of leukotrienes in neutrophils induced by opsonized *S*. *typhimurium* was significantly enhanced by ODN G4, and especially by G4 surrounded by phosphorothioate oligoguanosine runs (Figure 3). The synthesis of 5-LOX products reached the maximum even at 0.5 µM concentration of ODN, and effect of 1.0 µM ODNs was almost the same. For comparison, data are presented for a well characterized TLR9 antagonist ODN 4084-F [47] (Figure 3). The sum of LTB4, its isomers, and ω-OH-LTB4 were found to be produced to the greatest extent in the presence of ODNs G4-g_5_, g_2_-G4-g_5_, or 4084-F; the observed effects of all three oligonucleotides were close. Earlier, we have shown that TLR9 agonist CpG-ODN 2216, which also contain phosphorothioate fragments, but not a G-quadruplex-forming motif, induced the inhibition of leukotriene synthesis [44]. 

### 3.3. Effect of ODNs on Cytosol Ca^2+^ Level in PMNLs

Leukotriene synthesis is regulated via multiple mechanisms, and the induction of synthesis is tightly connected to the jump of cytosolic Ca^2+^ concentration [51,52]. It turned out that to one degree or another, each of the ODNs used in this study instantly caused an increase in the concentration of free calcium ions in the cytosol (Figure 4a), which undoubtedly indicates the stimulating impact of oligonucleotides. ODNs with terminal phosphorothioate oligoguanosines had a more pronounced effect on the influx of Ca^2+^ compared to the G4 containing only G-quadruplex-forming motif. We did not observe any significant impact of oligonucleotides on calcium ion raise caused by the interaction of neutrophils with opsonized bacteria. Explicitly affecting the synthesis of leukotrienes (Figure 3), TLR9 antagonist 4084-F, both by itself and in the presence of bacteria, unexpectedly had little effect on the intracellular signals of calcium ions (Figure 4b).

### 3.4. The Effect of ODNs on an Oxidative Burst in Neutrophils

In addition to the initiated calcium ion signaling, some peroxide tone of the cell is required for the full activation of 5-lipoxygenase [53]. Examining the oxidative status of neutrophils with cell-permeable H_2_DCF-DA revealed a significant increase in the fluorescence emission of the dye oxidation product when both ODNs with terminal modified oligoguanosine runs and TLR9 antagonist ODN 4084-F were added (Figure 5).

Elevated level of Ca^2+^ concentration, together with increased level of reactive oxygen species, contribute to stimulation of leukotriene synthesis by structured ODNs. To understand the mechanisms of oligonucleotide action on human neutrophils, we also examined the ability of ODNs under consideration to penetrate into cells. 

### 3.5. Ability of ODNs to Penetrate into the Cytoplasm of Cells

Most likely, differences in the strength of the effects of the studied ODNs are related to their bioavailability. Experiments with FAM-labeled ODNs also indicate a sharp increase in binding efficiency upon attachment of phosphorothioate oligoguanosine runs to the ends of telomeric G-quadruplex core (Figure 6a,b). A comparative assessment of total and intracellular fluorescence suggests that ODN G4 penetrates into the cell rather quickly, only a small amount of this oligonucleotide remains surface-bound. The presence of terminal modified oligoguanosines repeatedly increased surface binding, thereby contributing to the efficient uptake of G-quadruplex sequence motif into the cell. ODNs with oligoguanosines flanking G-quadruplex-forming sequence—G4-g_5_ and g_2_-G4-g_5_—exceeded the phosphorothioate TLR9 antagonist ODN 4084-F in the efficiency of binding to the cells (Figure 6b).

### 3.6. Folding Topology and Stability of G-quadruplexes Formed by Designed ODNs with Four Telomere Repeats Including End-Modified Analogues

The human telomeric G-quadruplex structures are known to be highly polymorphic due to different backbone orientations and type of internal loops. The prevalence of one particular conformation in solution is influenced by factors such as a number of G-rich repeats, the presence of terminal flanking nucleotides, counter-ion type and concentration, solvent composition, etc. [54]. Under in vitro conditions, G-quadruplexes adopt parallel, antiparallel, and hybrid (3+1) topologies characterized by different orientations of the four G-tracts (Figure 7a). Taking into account that cellular proteins recognize the specific G-quadruplex architecture [21,55,56], it was necessary to elucidate the folding topology and thermal stability of G-quadruplexes formed by ODNs under consideration (Table 1). All measurements were performed in buffer A containing potassium (5 mM) and sodium (140 mM) ions. This buffer solution was chosen to mimic the monovalent metal ion content in the extracellular space and in serum. A similar concentration of K^+^ ions (5.36 mM) was used for neutrophil incubation in HBSS medium. Moreover, a deliberately low K^+^ concentration was shown to be sufficient for G-quadruplex folding and the capture of the entire conformation transition in the accessible temperature range.

CD spectroscopy was used to detect the folding topology of G-quadruplexes formed by ODN G4 with four telomeric repeats (as a G-quadruplex-forming sequence motif), as well as by its derivatives with dangling oligoguanosines carrying modified phosphorothioate internucleotide bonds. This method allows discrimination of G-quadruplex structures with different arrangement of G-tracts. Figure 7b presents the CD spectra of the set of ODNs normalized to the same oligonucleotide concentration. G4 demonstrates the antiparallel topology, which is favored by telomeric TTAGGG repeats both in Na^+^ and K^+^ solutions [57]; it is characterized by the well-known marker bands: positive band around 290 nm followed by a negative one at 265 nm. Unexpectedly, the addition of phosphorothioate-containing oligoguanosines of varied lengths to the ends of G-quadruplex-forming motif causes sharp decrease of the positive CD peak at 295 nm and appearance of the strong positive peak at 262 nm reflecting the presence of a parallel-stranded G-quadruplex (Figure 7b). As evidenced from our previous data [26], the phosphorothioate bonds have a poor influence on the G-quadruplex structure. Therefore, the main reason of unusual folding pathways observed for oligonucleotides G4-g_5_ and g_2_-G4-g_5_ (Table 1) exposing signatures of both antiparallel and parallel G-quadruplex topologies is the extension of G4 at the ends with modified guanosine residues.

To better unveil the role of the terminal oligoguanosines in the G-quadruplex folding, UV melting profiles of ODNs G4, G4-g_5_, and g_2_-G4-g_5_ were measured under the same experimental conditions at 295 nm. The pattern of temperature dependence of UV absorbance at 295 nm is a marker of quadruplex structure. Unlike DNA duplex, whose melting is accompanied by a hyperchromic effect (usually at 260 nm), the G-quadruplex melting causes a decrease in the optical density of the sample, and this is a cooperative process [58]. The data obtained indicate a one-step melting of G-quadruplex formed by ODN G4, with Tm of 49 °C (Figure 7c). This observation suggests that G4 folds mainly into one conformation. In contrast, G4-g_5_ and g_2_-G4-g_5_ melting profiles display two different unfolding transitions confirming that at least two structures are present in the solution and participate in the melting process. The first transition with Tm (48 °C) almost coinciding with that of parent G4 (49 °C) is related to denaturing the antiparallel G-quadruplex structure, and the second transition with Tm value equals to 68 °C belongs to unfolding the parallel form. The fact that the hypochromic effect of G4 melting transition was significantly higher compared to that of G4-g_5_ and g_2_-G4-g_5_ is probably explained (i) by the presence of unstructured guanosine residues in combined ODNs, which absorb the UV light, but do not contribute to G-quadruplex unfolding, or (and) (ii) by reduced formation of G-quadruplex structure by G4-g_5_ and g_2_-G4-g_5_.

Thus, the melting properties of ODNs used for modulation of immune responses do not contradict CD spectroscopy data.

## 4. Discussion

Toll-like receptors sensing pathogens induce neutrophil activation and cause release of soluble mediators that can influence neutrophil’s functions. Since the activation of neutrophil TLRs is regulated by bacterial degradation products and genome DNA fragments released from the inflammatory region, TLRs become the perspective targets for synthetic ligands that mimic these products. So bacterial lipopolysaccharides via TLR4 receptors influence 5-LOX product synthesis in neutrophils [49] and macrophages [59,60]. DNA fragments act through TLR9 receptors, which are the most studied immune sensors of DNA [6]. 

Currently, G-rich ODNs capable of folding into non-canonical G-quadruplex structures are considered as promising molecular tools, which are recognized by a wide range of molecular targets such as proteins, viruses, and bacteria. In the majority of examples, unfolding the G-quadruplex structure leads to the loss of biological activity of G-rich ODNs. G-quadruplex DNA aptamers become interesting therapeutic and diagnostic alternatives to antibodies because of their high thermal stability, resistance to numerous serum nucleases, increased cellular uptake, and ease of chemical modification [61]. In the context of our work, it should be noted that DNA G-tracts are recognized by scavenger receptors, which are expressed on the surface of immune cells and can act as phagocytic receptors, as well as TLR co-receptors facilitating ODN uptake [62,63]. It has been shown synthetic ODNs consisting of a repeating telomere TTAGGG sequence can neutralize the immune activation induced by bacterial CpG-ODNs [11]. In our previous work, we demonstrated the ability of ODNs containing telomeric (TTAGGG)n sequence with different number of repeats, as well as their modified analogues, to influence the synthesis of 5-LOX-induced products in phagocyting neutrophils during their interaction with opsonized *S*. typhimurium. We demonstrated that only those ODNs that are able to fold into stable G-quadruplex structures significantly enhanced 5-LOX product formation [26]. Thus, the activation of leukotriene synthesis was completely lost when the formation of the G-quadruplex was prevented by replacing two guanines in G-tracts with adenine residues. It was also shown that mitochondrial and nuclear DNA from mammalian cells added to neutrophils were unable to induce neutrophil activation in vitro [64].

In this work, an attempt was made to determine what changes in primary and secondary structures of telomeric DNA repeats and their modified analogues would be crucial for the induction of functionally important neutrophil cellular responses. We examined a set of ODNs containing four human telomeric TTAGGG repeats including those with terminal modified oligoguanosines (Table 1). Oligoguanosine runs with phosphorothioate internucleotide bonds have been shown to significantly increase the cellular uptake of CpG-ODNs with phosphodiester backbone and to improve their immunostimulating activity in vitro and in vivo [39]. Here, we first attached the phosphorothioate oligoguanosines to the end(s) of telomere G-quadruplex motif and compared the potential of designed ODNs to regulate the main cellular responses: phagocytosis of opsonized *S*. *typhimurium* bacteria by human neutrophils, neutrophil adhesiveness (considered as frustrated phagocytosis), and phagocytosis-induced 5-LOX activation.

Flanking the telomeric G-quadruplex motif with modified oligoguanosines was revealed to improve the activation potential of ODNs tested. Both combined oligonucleotides (g_2_-G4-g_5_ and G4-g_5_) significantly enhanced phagocytic activity and adhesiveness of neutrophils compared to ODN with only a G-quadruplex motif (G4) and even to ODN 4084-F, which is a well-known TLR9 antagonist containing completely phosphorothioate backbone (Figure 1 and Figure 2), this effect being concentration dependent. We have shown that oligoguanosines attached to both the 3′- and 5′-ends of G4 motif improved ODN activity to a much greater extent than a single terminal modification. This regularity is not always observed for other ODN-mediated cellular responses. The pattern persisted when evaluating the ODN effect on the level of calcium ions in the cytosol (Figure 4); this indicates the ODN potential to enhance the synthesis of 5-LOX products. In contrast, ODN-induced synthesis of 5-LOX metabolites—LTB4 and its isomers—occurs approximately equally with ODNs G4-g_5_, g_2_-G4-g_5_ or 4084-F after neutrophil priming and further stimulation by opsonized *S*. *typhimurium* bacteria (Figure 3). The same is true also for ODN-mediated formation of intracellular oxidants that is coupled with increased leukotriene synthesis: ODNs G4-g_5_, g_2_-G4-g_5_, and 4084-F have about the same effect on the oxidative status of neutrophils (Figure 5). In any case, the ODNs containing telomeric G-quadruplex motif flanked by phosphorothioate oligoguanosines turned out more potent activators for key cellular responses compared to parent ODN (G4) lacking the dangling guanosines. Inhibitory ODN 4084-F is the shortest active 12-mer ODN that contains previously identified suppressive islets (CCT and GGG), appropriately spaced from each other (four nucleotides residues) and properly oriented in a single-stranded oligonucleotide (5’-CCT → GGG-3’) [48]. The biological activity of ODN 4084-F as inhibitor of CpG-induced immune activation has been proved [48]. The similarity in the behavior is an indirect confirmation of the similar biological activity of ODN 4084-F and ODN G4. Though we cannot exclude that some other biomolecules, not only TLR9, enable to mediate sequence-specific recognition of inhibitory ODNs.

The immune regulation using G-rich sequences is known to require relatively high ODN concentrations [65,66]. Usually, cellular uptake of ODNs is a crucial step for their interaction with a wide range of molecular targets because nucleic acids being polyanions cannot passively diffuse across cell membranes. According to our results, G4 sequences with terminal phosphorothioate oligoguanosines demonstrated crucially increased cellular uptake (Figure 6). Molecular mechanisms of synthetic ODNs’ internalization are not well understood. The most common is the hypothesis of clathrin-dependent endocytosis involving specific receptor proteins, e.g., stabilin receptors that bind phosphorothioate ODNs [67]. Furthermore, according to [68], intracellular trafficking and compartmentalization of phosphodiester ODNs, rather accumulating in the endosomes, differ significantly from those for their phosphorothioate analogs, significant part of which is found in cytosol. These data together with elevated resistance of structured modified DNA-based ligands to serum nucleases [69] could explain the more pronounced effects of combined ODNs G4-g_5_ and g_2_-G4-g_5_ on neutrophil’s cellular responses than the effect of ODN G4 with only a G-quadruplex motif. Our study has shown that besides cellular uptake neutrophil functions can also be modulated by G-quadruplex topology. It should be noted that just telomere G-quadruplexes could adopt a broad range of folding patterns [20,70].

Using spectroscopic techniques, we described the structural features of G-quadruplexes formed by ONDs under consideration at ionic conditions comparable to those found in the extracellular environment. As indicated by CD spectroscopy, ODN G4 folded into a G-quadruplex structure with antiparallel arrangement of G-tracts in water solution containing 140 mM NaCl and 5 mM KCl (Figure 7b); this conclusion did not contradict the literature data [57]. However, the CD spectra of ODNs G4-g_5_ and g_2_-G4-g_5_, displaying the spectral characteristics of both main G-quadruplex topologies, indicate that these oligonucleotides form a mixture of two different structures: the antiparallel G-quadruplex and the dominant parallel-stranded one (Figure 7b). Therefore, ODNs with four human telomeric repeats showed different conformational features in the presence/absence of oligoguanosines flanking the G-quadruplex core. UV spectroscopy data support this conclusion. While unfolding profile of ODN G4 demonstrates a clear single transition at 295 nm with Tm equals to 49 °C, UV melting of the ODNs G4-g_5_ and g_2_-G4-g_5_ displays two different transitions; Tm = 48° C is associated with the antiparallel G-quadruplex form, while Tm = 68 °C is related to more stable parallel topology. According to literature data, the addition of different nucleosides at the end of telomere DNA sequence is essential for the formation of different G-quadruplex structures [71]. Slow conversion of kinetically favored antiparallel G-quadruplex into thermodynamically favored parallel one was shown for the G-rich promoter region of *c*-*KIT* proto-oncogene [72].

The various conformational features of structured ODNs used in this study can make them selectively recognizable by TLR9 receptors. It is known that besides the thermodynamic stability, various G-quadruplex topologies have different conformation limits in terms of 5′- and 3′-end positions; parallel or hybrid (3+1) forms have the ends on opposite sides of G-quadruplex core and antiparallel ones on the same side. This might strongly affect the interaction of G-quadruplex ligands with biological targets. Thus, we assume that both the secondary structure of ODNs with human telomere TTAGGG repeats and their ability to penetrate into the cytoplasm of cells are important for inducing the neutrophil cellular effects. 

The data presented here show that ODNs with telomere-based sequences and their analogues are highly effective in neutrophil activation leading to increased formation of leukotrienes—soluble mediators that can potentiate microbial phagocytosis, enhance innate immunity, and influence other neutrophil’s functions. In particular, LTB4 has a short half-life in neutrophils and is rapidly oxidized into 20-OH-LTB4 and 20-COOH-LTB4 [73,74]. Earlier it was shown that inhibition of ω-oxidation of LTB4 increased LTB4 levels in vivo [75]. This negative feedback provided maximal accumulation of LTB4 at lower concentrations of ODNs, and the level of LTB4 did not increase with increasing of ODN concentration. 

The mechanism of the observed effects mediated by telomere-based ODNs on regulation of 5-LOX activity and leukotriene synthesis in human phagocyting neutrophils is not yet clear, but obviously involves cooperation of various receptors and signaling pathways. In our previous work, we described for the first time the role of folded exogenous G-quadruplex structures in neutrophil priming and activation of 5-LOX product synthesis [26]. Here we managed to find and characterize much more active G-quadruplex structures, the topology of which can be selectively changed in order to create synthetic structured DNAs with a serious potential in the manipulation with cellular receptors and in combating bacterial infection.

## 5. Conclusions

Multi-targeted effect of G-quadruplex-based aptamers described in [76] enable their interference with integral cellular processes, such as modulation of telomerase activity [56], proliferation of breast adenocarcinoma cells [76], suppression of inflammasome activity in macrophages [77]. In this study, we demonstrated that synthetic ODNs based on G-quadruplex-forming telomere repeats (d(TTAGGG)_4_) and their modified analogues significantly activate the leukotriene synthesis and other key cellular responses of human neutrophils. As revealed by spectroscopic techniques, ODN G4 and its derivatives with two combinations of dangling oligoguanosines attached to the ends of G4 fold into the stable G-quadruplex structures of different topologies. Just the flanking of the G-quadruplex-forming motif with oligoguanosine runs containing phosphorothioate backbone (i) dramatically affects the quadruplex folding pathways, leading to the preference for the parallel G-quadruplex topology instead of the antiparallel one found for the ODN G4, (ii) greatly increases the structure-dependent cellular uptake of ODNs, and (iii) significantly improves the activation of human neutrophils compared to the parent G4 lacking the dangling guanosines. Thus, G-quadruplex-forming sequences flanked with phosphorothioate oligoguanosines represent potential activators of phagocytosis and leukotriene synthesis in human neutrophils during their interaction with *S. typhimurium* bacteria. The results would be of benefit to the design and optimization of G-quadruplex-based anti-inflammatory agents.

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
