# Peer review of "The Potential of Telomeric G-Quadruplexes Containing Modified Oligoguanosine Overhangs in Activation of Bacterial Phagocytosis and Leukotriene Synthesis in Human Neutrophils"

_biomolecules, 2020, doi:10.3390/biom10020249_

Round 1

Reviewer 1 Report

This manuscript by G. F Sud'ina and co-workers examines the effect of a set of G-quadruplexes, that includes the telomeric sequence and variations modified with phosphorothioates, on mediating bacterial phagocytosis, cellular adhesion, production of leukotriene metabolites and 5-LOX, as well as calcium influx and production of ROS species in human neutrophils. The studies included appear systematic and convergent with regard to results.

The authors should address the following:

1) The introduction contains a lot of information on signaling pathways downstream of Toll-like receptors, directly relevant to the thesis of the manuscript. I find this to be in a language not easily followed by non-specialist readers. A visual representation (figure) of these pathways would be much preferred, and the English language should be edited as well by a native speaker to avoid any misconceptions.

2) On p.3, lines 3-5, the authors state: ".... which can be positive when ... promoting genome instability". This sentence should be rephrased - I find it very generic and perhaps misleading as the biological roles of G-quadruplexes are not yet fully elucidated, and certainly not all fit the description provided by this sentence.

3) The authors employ a trypan blue (TB) protocol to quench external/non-phagocytosed fluorescence from bacteria. Is this a standard method? If so, please provide reference.

4) In the isolation of 5-LOX metabolites, some additional information may be required on the separation of the desired metabolites from other compounds possibly contained in the same water/methanol extract and their positive identification.

5) On the UV spectroscopy melting curves, is there a reason why the authors monitor at 295 nm instead of the usual 260 nm?

6) In Figs. 1 and 2, please explain how you define 100% on the y-axis; also state more clearly what is the measurable property in each experiment.

7) In table 1, the addition of phosphorothioate oligo-g sequences to the telomeric sequence could, at least theoretically, create possibility for more than one G4 folds to co-exist, not due to alternative salt/buffer conditions but due to alternative sets of G-tracts participating in the G4 core. The existence of more than one population is, in fact, obvious in Fig. 7 from both UV melting (existence of both antiparallel and parallel), as well as the CD (small positive peak at 290 nm indicates an antiparallel population in addition to the dominant parallel). These do not necessarily employ the same G-tracts. It is not rationalized why the phosphorothioates needed to be oligo-g, but instead it is implied -without justification- that these do not participate in G4 core formation. I remain unconvinced that this is the case and would like the authors to address this point in detail.

8) In Fig. 3, the negative control no-bacteria AND no-ODNs is not included in the graphs, but it is mentioned that no 5-LOX products are detected in the absence of bacteria. This should be added to the graphs for comparison, if possible.

9) One of the experiments examined the cellular uptake of ODNs, which leads to the impression that the binding partner (not specified which!) is in the cytoplasm. However, in the introduction it is stated that ODNs interact with the Toll-like receptors on the cell surface to initiate downstream results. This needs to be clarified. Are we talking about dual mechanism of action? And are some of the observed results related to one but not the other? How do the pathways cross-talk? These are not obvious from the text!

10) In Fig 6(a), has any normalization been done so that the 2 graphs are comparable?

11) Even though it is not mentioned, it is implied that ODN 4084-F is not a G4. How do the authors explain that in several of the experiments this behaves similarly to the G4 structures examined?

12) On p. 14, in the discussion of CD spectra, the authors mention a "switch" of the antiparallel to parallel, when the sequence is replaced with the phosphorothioate one. I would urge the authors to refrain from using this term, as it creates a misconception that perhaps a change in conditions is what's causing the switch. This is not the case here, as we are dealing with 2 different sequences. The phosphorothioate sequence was never an antiparallel to switch to a parallel in the first place! It just folds from the beginning in a different conformation compared to the classic telomeric G4.

13) Finally in the melting temperature comparison of G4 (50 oC) and antiparallel g2-G4-g5 (48 oC), I feel the authors overstress this difference to make a case about phosphorothioates not participating the quadruplex core. I consider this to be an insignificant difference - even within statistical error - and would like some evidence that their claim is accurate.

By addressing these points, the quality of the manuscript can be improved significantly.

Reviewer 2 Report

This is an interesting paper reporting the effects of telomeric G-quadruplexes  on neutrophil phagocytosis and adhesion (and related phenomena). The work is generally well done employing a range of cell biological and chemical techniques. Given the potential importance of increasing neutrophil phagocytotic capacity to combatting antimicrobial resistance, this is an important contribution.

 Minor Point

The reported effect on Ca2+ signalling clearly need following up. The reported data are based on an unusual method. Ca2+ signals are usually transient and running  flow cytometry 20 mins after the event (even if cold) may have missed important events. 

Round 2

Reviewer 1 Report

With the changes introduced by the authors to reviewers' comments, the quality of the manuscript has improved.